# Seasonal Dynamics of Bacterial Community Structure in Diesel Oil-Contaminated Soil Cultivated with Tall Fescue (*Festuca arundinacea*)

**DOI:** 10.3390/ijerph19084629

**Published:** 2022-04-12

**Authors:** Yun-Yeong Lee, Soo Yeon Lee, Sang Don Lee, Kyung-Suk Cho

**Affiliations:** Department of Environmental Science and Engineering, Ewha Womans University, Seoul 03760, Korea; click529@ewhain.net (Y.-Y.L.); lsy0707@ewhain.net (S.Y.L.); lsd@ewha.ac.kr (S.D.L.)

**Keywords:** rhizoremediation, diesel-contaminated soil, tall fescue, bacterial community structures, association analysis

## Abstract

The objective of this study was to explore the seasonal characteristics of rhizoremediation and the bacterial community structure over the course of a year in soil contaminated with diesel oil. The soil was contaminated with diesel oil at a total petroleum hydrocarbon (TPH) concentration of 30,000 mg-TPH·kg-soil^−1^. Tall fescue seedlings were planted in the contaminated soil and rhizoremediation performance was monitored for 317 days. The TPH concentration gradually declined, reaching 75.6% after day 61. However, the TPH removability decreased by up to 30% after re-contamination in the fall and winter. The bacterial community structure exhibited distinct seasonal dynamics. Genus *Pseudomonas* significantly increased up to 55.7% in the winter, while the genera *Immundisolibacter* and *Lysobacter*, well-known petroleum hydrocarbon (PH)-degrading bacteria, were found to be positively linked to the TPH removal rate. Consequently, knowledge of this seasonal variation in rhizoremediation performance and the bacterial community structure is useful for the improvement of rhizoremediation in PH-contaminated environments.

## 1. Introduction

In recent decades, the increase in intensive petroleum production, transportation, and refinement has led to an increase in the accidental discharge of petroleum hydrocarbons (PHs) into the environment [1,2]. PH spills have a serious impact on the environment, especially wetlands [3]. Wetlands provide a variety of important ecosystem functions, including generating high primary productivity, removing excess nutrients from water, providing food and habitat for wildlife, and acting as a net sink for greenhouse gases [3]. In general, PHs persist for a longer time in wetlands than in unplanted sites due to the anaerobic conditions, thus the PH contamination of wetlands represents a more serious threat to the environment [2]. 

As a response to PH contamination, rhizoremediation is of particular use in wetlands, which are characterized by a diverse range of plants aerating the rhizosphere, potentially facilitating aerobic PH degradation [2]. Several plant species, such as longstamen rice (*Oryza longistaminata*), wild sorghum (*Sorghum arundinaceum*), giant thatching grass (*Hyparrhenia rufa*), club rush (*Scirpus grossus*), and black rush (*Juncus roemerianus*), have been employed for rhizoremediation in PH-contaminated wetlands [2,4,5]. In addition to these plant species, tall fescue (*Festuca arundinacea*), a perennial plant, is also useful for rhizoremediation due to its tolerance for pollutants and suitable root system [6]. Tall fescue grows well under a wide range of environmental stressors, including strongly acid to alkaline conditions, and its deep root system allows it to thrive in wetlands and other low-lying sites with high moisture [7]. Tall fescue is an introduced grass that is common in Pacific Northwest wetlands [8] and has been widely found in reclaimed tidal lands along with the mid-west coast of South Korea [9,10]. 

Rhizoremediation is affected by the interaction between plants and bacteria in the rhizosphere, with the bacterial community playing a significant role in the degradation of pollutants [11]. Accordingly, numerous studies have investigated rhizospheric bacterial communities concerning the rhizoremediation process at PH-contaminated sites. For example, Shen et al. (2018) assessed the bacterial community structure in the rhizosphere for *Agropyron cristatum* and *Cynodon dactylon* planted in petroleum-contaminated soil [12]. In addition, Hou et al. (2015) planted tall fescue in PH-contaminated soil and then added two plant-growth-promoting rhizobacteria strains, *Klebsiella* sp. D5A and *Pseudomonas* sp. SB [13]. After four months of rhizoremediation, the rhizospheric bacterial communities were compared. Guo et al. (2017) also compared the bacterial communities after 30 days of rhizoremediation using maize (*Zea mays* L.) and soybean (*Glycine max* L.) in polycyclic aromatic hydrocarbon (PAH)-contaminated soil [14]. Additionally, the rhizoremediation performance of tall fescue and the changes in the associated rhizospheric bacterial community in diesel-contaminated soil were evaluated for 85 days in our previous study [15]. However, these previous studies had been conducted for a relatively short period within three to four months. Because pollutants generally have greater persistence in wetlands [2], the changes in pollutant levels and bacterial communities need to be monitored over a longer period at contaminated sites. Therefore, in this study, seasonal PH remediation performance was investigated over 317 days using tall fescue, which has the potential to be employed for rhizoremediation in wetlands and is suitable for long-term experiments. Furthermore, the seasonal dynamics of the bacterial community structure in the tall fescue rhizosphere were analyzed using next-generation sequencing (NGS) over the experimental period. Finally, relationships between rhizoremediation performance, environmental factors, and the bacterial community were comprehensively investigated. 

## 2. Materials and Methods

### 2.1. Soil Preparation

Soil was taken from a site near Ewha Womans University in Seoul, South Korea (37°57′ N, 126°95′ E). Weeds and stones were removed from the soil, which was then mixed with compost at a 9.5:0.5 ratio (*w*/*w*). The compost, which was purchased from a commercial vendor (Seokgang Green Fertilizer Inc., Icheon, Korea) and used as a soil amendment to enhance rhizoremediation performance, was composed of swine manure (40%), sawdust (49%), cow manure (10%), and bacterial inoculum (1%). The soil was then artificially contaminated with diesel oil at a total petroleum hydrocarbon (TPH) concentration of 30,000 mg-TPH·kg-soil^−1^.

### 2.2. Experimental Setup

The bottom of a pot (60 cm × 120 cm × 20 cm; L × W × H) was evenly lined with 5 L of perlite (Kyungdong One Co., Ltd., Seoul, Korea). Contaminated soil (180 kg) was then added to the pot. Tall fescue seeds were sown in the garden on the rooftop of the New Engineering Building, Ewha Womans University, and cultivated for 2 months. After 2 months, 100 tall fescue seedlings were planted in the pot. The pot experiment was conducted for 317 days (27 May 2020–9 April 2021) to investigate the seasonal rhizoremediation performance. Soil samples were collected at a depth of 5–15 cm nearby tall fescue roots. Soil samples were randomly sampled from the pot once a month from May to October, and once every two months during the winter season (from October to April). The sampling was conducted on days 0 (27 May 2020), 33 (29 June 2020), 61 (27 July 2020), 84 (19 August 2020), 117 (21 September 2020), 152 (26 October 2020), 210 (23 December 2020), 271 (22 February 2021), and 317 (9 April 2021). 

The collected soil samples were sieved through a 2-mm mesh. The pH and water content of the soil was measured according to the Korean Standard Soil Analysis Method (ES.07302.1b and ES.07301.1b) [16]. The organic content was measured based on the Korean Standard Waste Analysis Method (ES.06303.1c) [17]. Air temperature and daily precipitation in Seoul were obtained from the Korea Meteorological Administration (https://data.kma.go.kr/, accessed on 9 April 2021).

### 2.3. Instrumental Analysis

The residual TPH concentration was measured to monitor the diesel oil concentration in the contaminated soil. Freeze-dried soil samples (1.5 g) were mixed with 10 mL of hexane-acetone (1:1, *v*/*v*) solution and then incubated at 30 °C and 200 rpm for 30 min. After incubation, the mixture was left at room temperature for 5 min to obtain the supernatant. The TPH concentration extracted from the supernatant was measured using a gas chromatograph (6890N, Agilent Technologies Inc., Santa Clara, CA, USA) equipped with a flame ionization detector (Agilent Technologies Inc.). The operating conditions for the gas chromatography system are described in our previous study [15].

### 2.4. Bacterial Community Structure Analysis

Sieved soil samples (0.3 g) were added to a sterilized 1.5-mL micro-centrifuge tube and stored at −23 °C before DNA extraction. Genomic DNA was extracted using a NucleoSpin^®^ Soil Kit (Macherey-Nagel GmbH, Düren, Germany) following the manufacturer’s protocol and FastPret-24^TM^ (MP Biomedials, Irvine, CA, USA) was used for bead beating. The genomic DNA samples were eluted in 50 μL of elution buffer and quantified using a SpectraMax QuickDrop Spectrophotometer (Molecular Devices, San Jose, CA, USA). The extracted genomic DNA samples were stored at −23 °C before use.

The seasonal dynamics of the bacterial community in the tall fescue rhizosphere were also analyzed. Dual-index PCR was performed based on a primer set of 515F (5′-TGC CAG CMG CCG CGG TAA-3′) and 806R (5′-GGA CTA CHV GGG TWT CTA AT-3′), which targets the V4 region of the 16S rRNA gene [18,19]. Details of this process are described in our previous report [15]. All of the purified samples in sterilized 1.5 mL micro-centrifuge tubes were analyzed by Macrogen Inc. (Seoul, Korea) using the Illumina Miseq sequencing platform (Illumina Inc., San Diego, CA, USA). The sequence reads were processed by Macrogen Inc. using QIIME 1.9, as described in a previous report [15]. The obtained sequencing reads were deposited in the National Center for Biotechnology Information (NCBI) Sequence Read Archive (http://www.ncbi.nlm.nih.gov, accessed on 26 December 2021) under accession number SRP352503.

### 2.5. Statistical Analysis

The change in bacterial community structure over the period was assessed using principal component analysis (PCA) and a hierarchically clustered heat map. PCA was conducted using UniFrac and CANOCO 4.5 software (Microcomputer Power, Ithaca, NY, USA) [20]. The heat map for the bacterial community was visualized using the gplots package in R (v 4.0.1). The correlations of rhizoremediation performance, environmental factors, and bacterial genera were described using extended local similarity analysis (eLSA) and visualized with Cytoscape 3.7.1 [21,22]. One-way ANOVA followed by multiple comparison tests were conducted using R software (v 4.0.1) with a 0.05 *p*-value significance threshold.

## 3. Results and Discussion

### 3.1. Seasonal Degradation Performance of TPH in Diesel-Contaminated Soil

Figure 1a,b show the seasonal air temperature and daily precipitation in Seoul, Korea. We divided the experimental period into four sub-periods based on the average ambient temperature: summer (20–30 °C, 27 May–19 September), autumn (5–20 °C, 20 September–19 November), winter (below 5 °C, 20 November–28 February), and spring (5–20 °C, 1 May–9 April). The average ambient temperature for each period was 24.2 °C for summer, 14.3 °C for autumn, 0.2 °C for winter, and 10.2 °C for spring (Figure 1a). Daily precipitation was most frequent in summer, with an average of 16.8 mm and a maximum of 103.1 mm (Figure 1b). On the other hand, the precipitation was relatively light from autumn to spring. The average and maximum precipitation was 12.8 mm and 86 mm, respectively, in autumn, 1.1 mm and 9.2 mm in winter, and 12.9 mm and 67.5 mm in spring.

The physicochemical properties of the soil are summarized in Figure 1c–e. The pH remained constant over the study period with an average of 6.8 (Figure 1c). The water content in soil was affected by the daily precipitation, with an average of 40.8%, a minimum of 33.4%, and a maximum of 47.3% in summer (Figure 1d). On the other hand, autumn to spring had an average of 28.2%, a minimum of 18.9%, and a maximum of 32.5% (Figure 1d). The organic matter levels gradually decreased from an initial value of 11.0% to a final value of 8.8% (Figure 1e).

Figure 2 presents the residual TPH concentration in the soil and the removal efficiency over the experimental period. The initial TPH concentration was 24,447 mg-TPH·kg-soil^−1^ on day 0. It decreased to 15,416 mg-TPH·kg-soil^−1^ on day 33, representing a removal efficiency of 36.9%. By day 61, the TPH concentration had greatly decreased to 7050 mg-TPH·kg-soil^−1^, a removal efficiency of 71.2% (*p* < 0.05). The TPH concentration did not change significantly after day 84. The average concentration and removal efficiency was 5970–7535 mg-TPH·kg-soil^−1^ and 69.2–75.6%, respectively, from days 84 to 152 (*p* > 0.05). Accordingly, it was assumed that the residual TPH was no longer being degraded in the soil, thus the soil was re-contaminated with diesel oil at a concentration of 41,873 mg-TPH·kg-soil^−1^ on day 153. On day 210, 17.9% of the TPH had been removed, leaving an average concentration of 34,371 mg-TPH·kg-soil^−1^. After the re-contamination, the TPH removal efficiency was not as efficient as it was before re-contamination, with the average concentration and removal efficiency being 26,154–29,259 mg-TPH·kg-soil^−1^ and 30.1–37.5%, respectively, from days 271 to 317 (*p* > 0.05).

Tall fescue is a rhizoremediation species that is widely used to treat PH-contaminated soils because they tolerate pollutants and a fibrous root system [6]. However, there has been little previous research on the use of tall fescue for rhizoremediation in wetland environments. There have been a number of studies that have explored rhizoremediation in wetland environments under different experimental conditions and with different plant species. For example, Al-Baldawi et al. (2013) employed *Scirpus grossus* for rhizoremediation in a constructed wetland and the highest TPH removal rate was 84.1–88.3% [23]. In addition, Lin and Mendelssohn (2009) investigated rhizoremediation performance using *Juncus roemerianus* in marsh sediment contaminated with diesel fuel, and it was reported that more than 50% of diesel fuel was removed after 12 months [2]. In another study, the rhizoremediation of TPH contamination was evaluated using four plants (*Oryza longistaminata*, *Sorghum arundinaceum*, *Tithonia diversifolia*, and *Hyparrhenia rufa*) in a wetland in South Sudan, with a removal rate of 50–74% [4].

Similar to these results, the TPH removal rate in the present study ranged from 69.2% to 75.6% for days 61–152. However, after re-contamination with diesel on day 153, the TPH removal rate declined to 30.1–37.5%. This observation might be because the bacterial activity for TPH degradation in the tall fescue rhizosphere was disrupted as the temperature fell after the re-contamination. Temperature is one of the important factors affecting biodegradation of PH in soil [24]. It is generally reported that excessively high or low temperatures inhibit bacterial metabolism for PH degradation in contaminated soils [24]. Increase in temperature improves solubility and diffusion of contaminants, while low temperature decreases them which delays onset of biodegradation [25]. According to previous reports, the optimum temperature ranged from 30 to 35 °C for the bacterial degradation of PH [15,26]. Yeung et al. (1997) stated that hydrocarbon degradation rate dropped rapidly at low temperature (particularly below 10 °C) [26].

Moreover, the increase in the TPH concentration in the soil due to re-contamination may decrease TPH removal. According to previous reports, the TPH removal rate decreases as the initial TPH concentration increases, which was consistent with the result in this study [6,15,27]. Particularly, our previous report compared TPH removal at different initial TPH concentrations during rhizoremediation using tall fescue [15], with the TPH removal efficiency reaching 50.0–67.2% when the initial concentration was less than 30,000 mg-TPH·kg soil^−1^ and falling dramatically to 36.5% when the initial TPH concentration was 40,000 mg-TPH·kg soil^−1^. Consequently, it is thought that rhizoremediation was hindered by the threshold for TPH degradation being exceeded after re-contamination.

Decrease in soil water content diminishes PH biodegradation rate due to an inadequate supply of water for sustain metabolism [28,29]. Lee et al. (2018) reported that the optimum moisture content for PH biodegradation was found to be at 30 to 80% [28]. On the other hand, Haghollahi et al. (2016) investigated PH biodegradation with different soil water contents of 10% and 20%, but no significant difference was found as the soil water content [29]. They reported that the soil texture had a more significant effect on PH degradation than soil water content, with the TPH removal rates in sandy soils significantly higher than clay and coarse soils [29]. In a study by Kogbara et al. (2015), PH degradation was studied using five types of soil (sand, loamy sand, sandy loam, silt clay, and clay), showing the highest PH degradation in sandy loam soil [30]. Similarly, the soil texture used in this study was sandy loam as described in our previous report [15]. In addition, the water content of soil was maintained in a range of 25.9 to 32.5% after re-contamination in this study. Consequently, even if soil water content is a factor that considerably affect the PH biodegradation, it is not expected that it had a significant effect on the TPH reduction after re-contamination in this study.

### 3.2. Seasonal Dynamics of the Bacterial Community Structure

The seasonal dynamics of the bacterial community were analyzed with an Illumina MiSeq sequencing platform. The operational taxonomic units (OTUs) and alpha diversity indices are presented in Table 1. The number of OTUs ranged between 538 and 1383 during the experimental period, with the highest number of OTUs and highest alpha diversity recorded on day 84 (19 August). The Chao1 and Shannon indices, which are used to measure species richness and evenness, respectively, were 752.1–1698.5 and 3.41–7.02, respectively, over the entire period. The inverse Simpson index ranged from 0.71 to 0.98, and the Good’s coverage ranged from 0.987 to 1.000.

Figure 3 presents the changes in the rhizospheric bacterial community structure over the experimental period, with a focus on those genera with a relative abundance of at least 1%. At the phylum level, *Gamma-proteobacteria* (40.1–84.6%) dominated in the community (Figure 3a), followed by *Alpha-proteobacteria* (0.7–26.7%) and *Beta-proteobacteria* (9.9–17.9%) (Figure 3b). Other phylum such as *Acidobacter*, *Bacteroidetes*, and *Firmicutes* were 4.8–25.7% in the bacterial community (Figure 3c). In the initial soil sample on day 0, which was not greatly affected by diesel oil contamination, *Acinetobacter* (62.5%) was most dominant, followed by *Pseudomonaas* (10.2%). As shown in Figure 3, a considerable difference in the bacterial community structure was observed on day 33, with *Pseudomonas* exhibiting the highest abundance (17.3%), followed by *Aquabacterium* (8.7%), *Solimonas* (8.1%), and *Immundisolibacter* (7.8%). The bacterial community structure was similar for days 61–117, with *Immundisolibacter* (9.5–12.5%), *Umboniibacter* (9.9–13.3%), and *Sphingomonas* (6.6–10.5%) generally abundant. On the other hand, the bacterial community structure was slightly different on day 152, when it was dominated by *Marinobacter* (12.7%) and *Sphingomonas* (10.9%). On day 210, which was during winter, the relative abundance of *Pseudomonas* increased to 15.7%, followed by *Sphingomonas* (11.9%) and *Massilia* (9.5%). On day 271, the relative abundance of *Pseudomonas* increased considerably to 55.7%, whereas the other genera were all below 8%. *Pseudomonas* remained dominant on day 317 at 26.9%, followed by *Sphingomonas* (9.0%). 

The changes in the bacterial community structure were analyzed using PCA and a heat map (Figure 4). The bacterial community on day 0 was highly distinguishable from the other sampling points (Figure 4a), with *Acinetobacter* having the highest relative abundance (Figure 4b). On day 33, the bacterial community structure started to exhibit differences from day 0. In particular, the community structure in summer and autumn (days 33–152) was distinguishable from that in winter and spring (days 210–317). The hierarchically clustered heat map revealed that the genera *Sphingomonas*, *Umboniibacter*, and *Immundisolibacter* accounted for a relatively high abundance in summer and autumn, while *Pseudomonas* and *Sphingomonas* were more dominant in winter and spring (Figure 4b).

In this study, *Acinetobacter* had a very high relative abundance of 62.5% on day 0. *Acinetobacter* is a strictly aerobic genus and can be found in a variety of natural environments, including soil, fresh water, oceans, and sediments [31]. Kim et al. (2008) and Krizova et al. (2014) isolated *Acinetobacter* from forest soil and water [32,33], while it has also been isolated from hydrocarbon-contaminated sites, demonstrating the ability to degrade hydrocarbons [31]. According to a report by Chen et al. (2012), *Acinetobacter* sp. isolated from PH-contaminated soil produces biosurfactant for PH degradation [34]. Thus, *Acinetobacter* may have influenced TPH degradation in the initial stages of the present study.

During the rhizoremediation process, *Immundisolibacter*, *Sphingomonas*, and *Solimonas* became more abundant from day 33. These genera are PH-degrading bacteria that are regularly found at PH-polluted sites. Corteselli et al. (2017) isolated *Immundisolibacter* strain TR3.2^T^ from an aerobic bioreactor used to treat PAH-contaminated soil, and this strain grew well in a PAH-amended medium as the sole carbon source [35]. In other studies, *Immundisolibacter* has been found in oil-polluted soil in the coastal area of Shandong, China [36] and heavily hydrocarbon-polluted soil in Grabownica, Poland [37]. *Sphingomonas* has been reported to have the ability to degrade PAH [38,39]. Eight strains of *Sphingomonas* were isolated from petroleum-contaminated soil in Shenfu, China [39], six of which had degradation rates of 63.2–87.2% for fluorine. Two of the strains also exhibited a degradation rate of 60.6–64.8% for phenanthrene. In other studies, *Solimonas* was found in soil contaminated with hydrocarbons and metals [40] and in the rhizosphere of grasses in petroleum-contaminated soils [41]. In both studies, *Solimonas* was shown to contain the gene *alk*B, which encodes alkane monooxygenase for alkane degradation.

The relative abundance of *Pseudomonas* increased in the winter (after day 210), reaching a maximum of 55.7%. According to previous studies, *Pseudomonas* grows well and can maintain its metabolic activity at temperatures below 4 °C [42,43]. In addition, *Pseudomonas* sp. has previously shown promise for use in the bioremediation of oil-contaminated soils at low temperatures. For example, Stallwood et al. (2005) isolated *Pseudomonas* strain ST41 from oil-contaminated and pristine soils in Antarctica and investigated its ability to degrade oil at 4 °C [44]. They demonstrated that strain ST41 is a hydrocarbon-degrading bacterium that can utilize marine gas oil as its sole carbon and energy source. In another study, *Pseudomonas* strain JM2 was isolated from the sewage sludge of a wastewater treatment facility in northeastern China, where the mean annual temperature is 6 °C [45] and degraded fluorine by 24% and phenanthrene by 12% at 4 °C. In addition, Ma et al. (2006) isolated 22 strains from oil-contaminated Antarctic soil and cultured them in a medium containing PAH [46]. Of these, 21 strains had high homology with *Pseudomonas*, and all showed high PAH degradation efficiency at 4 °C. In general, viscosity of PH increases at low temperatures, resulting in inhibition of biodegradation [25]. Therefore, production of biosurfactants or bio-emulsifiers plays an significant role in PH degradation at low temperature [25]. *Pseudomonas* is capable of producing rhamnolipid-type biosurfactants [47], and Malavenda et al. (2015) reported the production of biosurfactants by *Pseudomonas* at low temperature (4–15 °C) [48]. These results show that *Pseudomonas* has the potential to be applied to bioremediation during the winter or in cold regions, thus *Pseudomonas* may have influenced the degradation of diesel oil during winter in the present study.

### 3.3. Extended Local Similarity Analysis (eLSA)

Association network analysis was conducted using eLSA to evaluate the relationship between rhizoremediation, the environmental conditions, and the bacterial genera. The blue solid and red dashed edges in Figure 5 represent positive and negative relationships, respectively. There were positive relationships between most genera. Ambient temperature had a direct negative relationship with *Bradyrhizobium* and *Massilia*, while the water content of the soil was also negatively related to *Massilia*. *Bradyrhizobium* is a rhizospheric genus [49], and previous studies have demonstrated that several *Bradyrhizobium* species are dominant in soil at low temperatures, promoting soybean growth in cold areas [50,51]. It has also been shown that *Massilia* can degrade several hydrocarbons such as PAH and benzene, toluene, ethylbenzene, and xylene (BTEX) [52,53], while some *Massilia* species have been isolated from ice core or arctic soils and are tolerant to low temperatures [53,54].

In the present study, TPH removal efficiency had a positive direct correlation with *Immundisolibacter* and *Lysobacter*. As described above, *Immundisolibacter* contains PH-degrading bacteria, while *Lysobacter* is part of the γ-proteobacteria phylum and is commonly found in natural environments such as soil and fresh water. According to [55]. *Lysobacter* has been reported to have various ecological functions, including the promotion of plant growth, thus it is widely found in the rhizosphere of a variety of plants, such as pepper, strawberry, rice, tomato, and soybean. The same authors also reported that *Lysobacter* has the ability to decompose soil pollutants, thus it has attracted attention for its potential role in the bioremediation of contaminated sites. For example, *Lysobacter* was the most abundant bacterial genera during the bioremediation of PH-contaminated soils [13,56], and *Laysobacter* isolated from a marine environment was demonstrated to be able to degrade TPH and PAH by producing a biosurfactant [57].

## 4. Conclusions

Rhizoremediation, which is a result of the synergy between plants and rhizospheric bacteria, can aid in the restoration of contaminated environments, including wetlands. Rhizoremediation is a sustainable and cost-effective approach for PHs-contaminated soil restoration because it does not require additional chemicals amendment. Tall fescue has the potential to be employed in the rhizoremediation of contaminated wetland, where pollutants have greater persistence, because tall fescue is a perennial that is tolerant to environmental stressors. In the present study, seasonal rhizoremediation performance was investigated using tall fescue in diesel oil-contaminated soil over a 317-day study period. After the soil was contaminated with diesel oil at an initial concentration of 24,447 mg-TPH·kg dry soil^−1^, the residual TPH concentration gradually decreased, reaching a maximum removal efficiency of 75.6%. The seasonal dynamics of the bacterial community in the rhizosphere were also monitored using Illumina Miseq sequencing analysis. The bacterial community structure changed over the study period. On day 0, genus *Acinetobacter* (62.5%) was dominant, while *Immundisolibacter*, *Sphingomonas*, and *Solimonas* were dominant during the rhizoremediation process. The relative abundance of *Pseudomonas* also greatly increased to reach 55.7% in the winter. Association network analysis revealed that the TPH removal efficiency had a direct positive relationship with *Immundisolibacter* and *Lysobacter*, which are well-known PH-degrading bacteria. This study has limitations in measuring plant biomass and soil enzyme activities, which play a significant role in rhizoremediation. Therefore, it is required to be supplemented in subsequent studies. Nevertheless, this study investigated the seasonal changes in diesel oil pollution levels and the bacterial community structure in detail. The rhizospheric bacterial community was found to play an important role in rhizoremediation. Therefore, the results obtained in this study can provide a basis for the improvement of the rhizoremediation of PH-contaminated sites.

## Figures and Tables

**Figure 1 ijerph-19-04629-f001:**
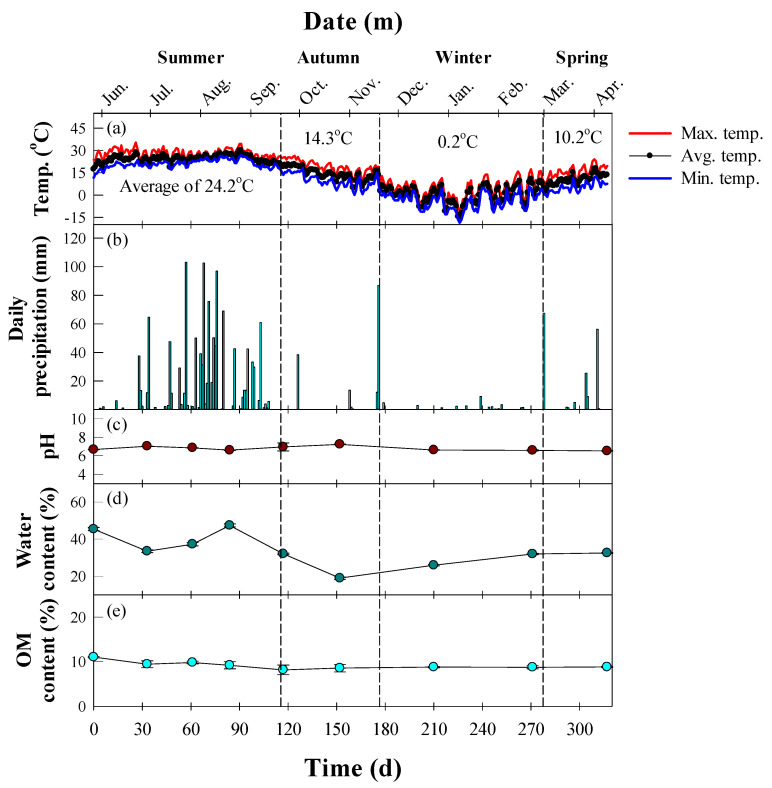
Time profile of the weather in Seoul, South Korea, and the physicochemical properties of the diesel-contaminated soil. (**a**) Ambient temperature and (**b**) daily precipitation in Seoul over the experimental period. (**c**) pH, (**d**) water content (%), and (**e**) organic matter content (%) of the soil.

**Figure 2 ijerph-19-04629-f002:**
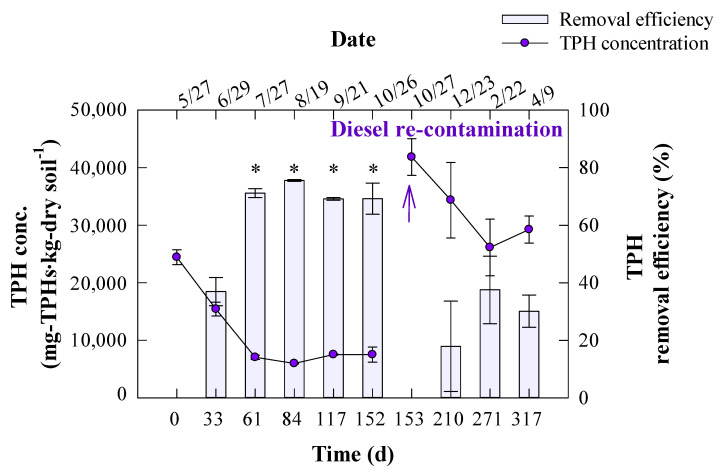
Residual TPH concentration and removal efficiency in the diesel-contaminated soil. Asterisks indicate significant differences (*p* < 0.05).

**Figure 3 ijerph-19-04629-f003:**
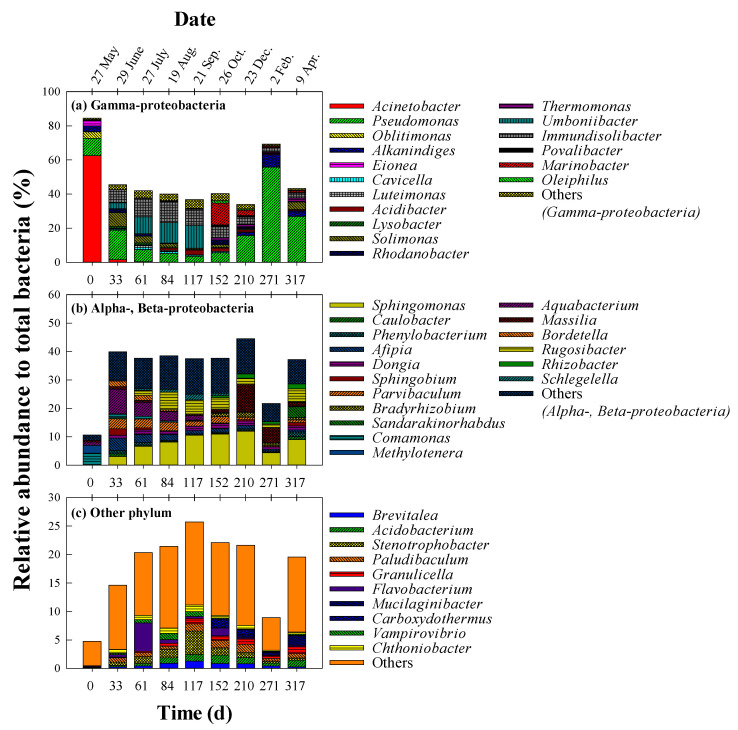
Changes in the bacterial community structure in the diesel-contaminated soil. (**a**) Genera of Gamma-proteobacteria, (**b**) Alpha- and Beta-proteobacteria, and (**c**) other phylum. Only genera with a relative abundance of >1% are illustrated.

**Figure 4 ijerph-19-04629-f004:**
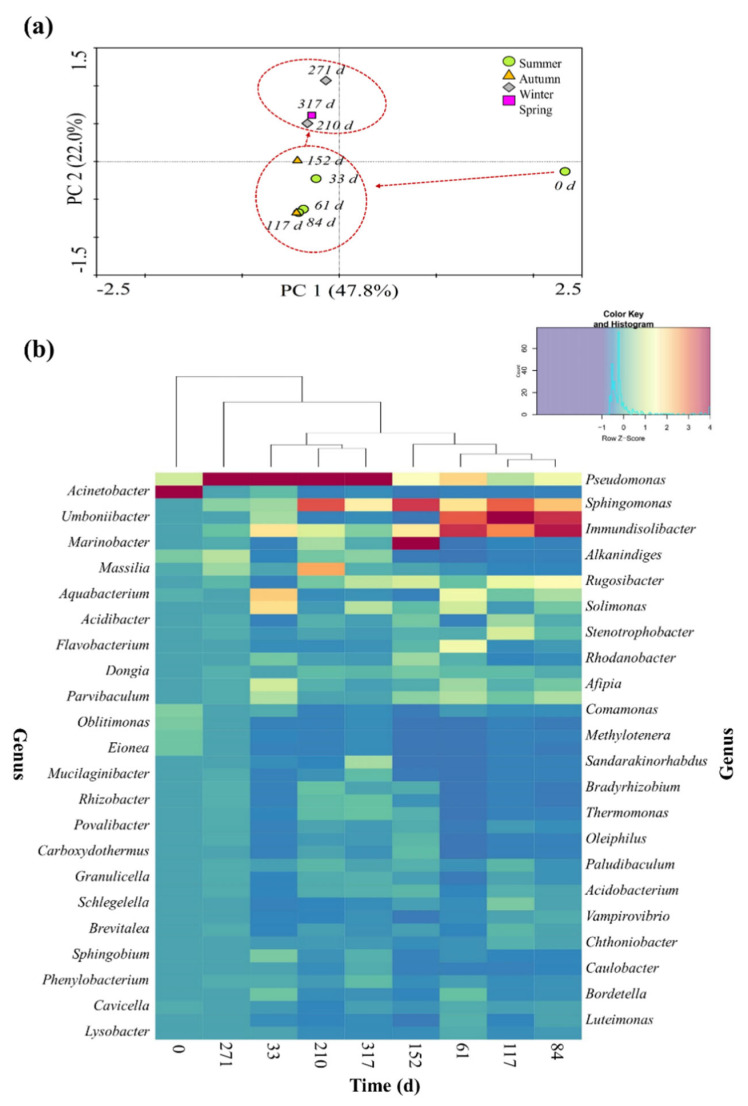
Seasonal changes in the soil bacterial community. (**a**) Principal component analysis (PCA) of the bacterial community structure in the soil. (**b**) Hierarchically clustered heat map of the bacterial distribution at the genus level. The rows represent different soil samples and the columns indicate the relative abundance of each bacterial group. The relative abundance is denoted by the intensity of the color.

**Figure 5 ijerph-19-04629-f005:**
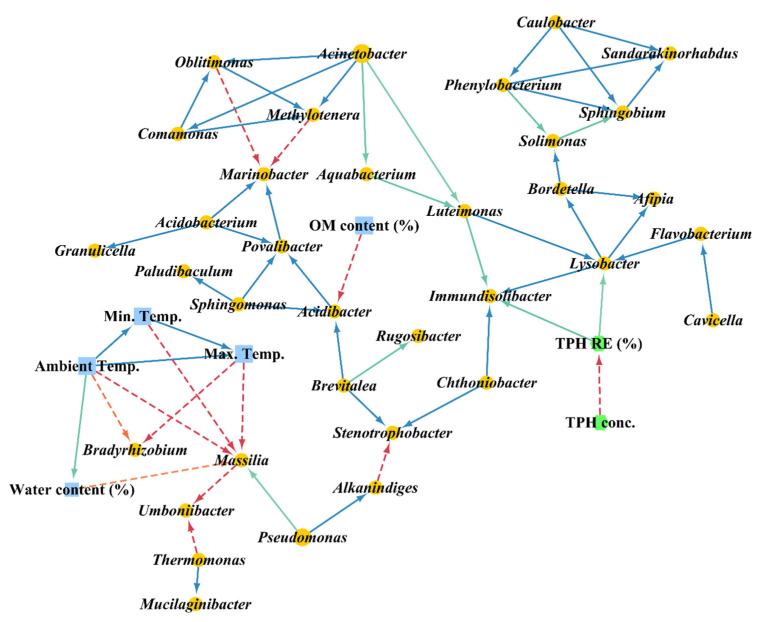
Association network analysis between the environmental conditions, pollutant concentrations, and bacterial genera. Solid (blue and green) and dashed (red) edges indicate positive and negative relationships, respectively (*p* < 0.005).

**Table 1 ijerph-19-04629-t001:** Operational taxonomic units (OTUs) and alpha-diversity indices for the bacterial soil community.

Date	Sampling Time (d)	OTUs	Chao1 ^a^	Shannon ^b^	Inverse Simpson ^c^	Good’s Coverage ^d^
27 May	0	565	756.4	3.41	0.71	0.995
29 June	33	970	1380.7	6.39	0.96	0.987
27 July	61	538	830.3	5.20	0.89	0.972
19 August	84	1383	1698.5	7.02	0.97	0.993
21 September	117	901	1163.3	5.62	0.94	0.995
26 October	152	788	794.9	7.58	0.98	1.000
23 December	210	729	740.8	6.62	0.96	0.999
22 February	271	744	752.1	6.71	0.95	0.999
9 April	317	749	757.0	7.01	0.97	0.999

^a^ The Chao1 index measures the bacterial population richness. ^b^ The Shannon index measures the number and evenness of the species. ^c^ Inverse Simpson index represents the probability that two randomly selected individuals in the habitat will belong to the same species. It is calculated as D = 1 − [Σ*n*(*n* − 1)/*N*(*N* − 1)], where *n* is the number of individuals of each species and *N* is the total number of individuals of all species. ^d^ Good coverage gives a relative measure of how well the sample represents the larger environment. It is calculated as C = 1 − (*s* − *n*), where *s* is the number of unique OTUs and *n* is the number of individuals in the sample.

## Data Availability

Not applicable.

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
