# Peer review of "Seasonal Dynamics of Bacterial Community Structure in Diesel Oil-Contaminated Soil Cultivated with Tall Fescue (Festuca arundinacea)"

_ijerph, 2022, doi:10.3390/ijerph19084629_

Round 1
Reviewer 1 Report
The graphics need to be improved, in particular Figure 3 and Figure 4b for clarity and understanding.
The introduction could benefit from an additional figure to introduce the statistical importance of the work, but this is option as I found the introduction very easy and straightforward to read.
I have attached minor comments on the English language but overall, the manuscript was very well written and easy to understand.

Author Response
- Reviewer #1
The graphics need to be improved, in particular Figure 3 and Figure 4b for clarity and understanding. The introduction could benefit from an additional figure to introduce the statistical importance of the work, but this is option as I found the introduction very easy and straightforward to read. I have attached minor comments on the English language but overall, the manuscript was very well written and easy to understand.
=> We have much appreciated this valuable comment. We have improved the graphics (Figures 3 and 4b) as the recommend. Figure 3 has been separated into different three graphs by the phylum level. The genera in Figure 4 has been changed to make them easy to read. Also, we have checked the English writing of the manuscript as the attached minor comments.
Page 1, line 20 in Abstract;
Page 1, line 32 in Highlights;
Page 2, lines 54 & 59;
Page 3, line 106;
Page 6, line 190;
Page 6, line 204;
Pages 8–9, lines 259–262; 277–278 & Figure 3;
Page 9, Figure 4.
Reviewer 2 Report
Rejected
Author Response
rejected
Reviewer 3 Report
The manuscript entitled “Seasonal dynamics of bacterial community structure in diesel oil-contaminated soil cultivated with tall fescue (Festuca arun-dinacea)” authored by Yun-Yeong Lee, Soo Yeon Lee, Sang Don Lee and Kyung-Suk Cho has been reviewed.
This paper investigates the cyclic characteristics of rhizoremediation in Petroleum PH-contaminated wetlands and the role of petroleum hydrocarbon (PH)-degrading bacteria such as Genus Pseudomonas, genera Immundisolibacter and Lysobacter studied in soil contaminated with diesel oil. Tall fescue seedlings were planted in the contaminated soil and rhizoremediation performance was monitored for 317 days. The scientific content of the manuscript is very sensational and significant for the bioremediation of soil pollutants and rejuvenation of the PH-contaminated sites using various bacterial genera. The paper reports the solution for a contemporary issue and worth publishable. The manuscript is very well organized and written with ample references and adequate for publication after making some minor corrections described below.
Q1. P2, L-50-55, “Tall fescue grows well under a wide range of environmental stressors, including strongly acid to alkaline conditions, and its deep root system allows it to thrive in wetlands and other low-lying sites with high moisture Tall fescue is an introduced grass that is common in Pacific Northwest wetlands and has been widely found in reclaimed tidal lands along the mid-west coast of South Korea”.
Please correct the English grammar.
Q2. P2, L-78-79, “Finally, the association between rhizoremediation performance, environmental factors, and the bacterial community was comprehensively assessed.”
Please correct the grammar of the sentence
Q3. P4, L-139-141, “The association between rhizoremediation performance, environmental factors, and bacterial genera was analyzed using extended local similarity analysis (eLSA) and visualized with Cytoscape”
Please correct the English grammar
Q4. P4, L-170-172, “The TPH concentration subsequently did not change significantly, with an average concentration and removal efficiency of 5,970–7,535 mg-TPH·kg-soil-1 and 69.2–75.6%, respectively, from days 84 to 152.”
Please correct the sentence.
Q5. P6, L-195-199, “Similar to these results, the TPH removal rate in the present study ranged from 69.2% to 75.6% for days 61–152. However, after re-contamination with diesel on day 153, the TPH removal rate declined to 30.1–37.5%. This might be because the bacterial activity for TPH degradation in the tall fescue rhizosphere was disrupted as the temperature fell after the re-contamination”.
Here the author claims that after re-contamination with diesel, the bacterial activity for TPH degradation in the tall fescue rhizosphere was disrupted. What are the other plausible reasons for the declination of TPH removal rate? Is this happen only due to temperature? What about water content and organic matter content? Also please correct the sentence structure.
Q6. P7, L-238-241, “On day 210, which was during winter, the relative abundance of Pseudomonas increased to 15.7%, followed by Sphingomonas (11.9%) and Massilia (9.5%). On day 271, the relative abundance of Pseudomonas increased considerably to 55.7%, whereas the other genera were all below 8%”.
According to the experimental procedure narrated in Page 3; L99, the day 210 (December 23rd, 2020) and day 271 (February 22nd, 2021), which is in the peak of winter and the data shows that On day 271, the relative abundance of Pseudomonas increased considerably to 55.7%. Could you please explain why this is happened to only Pseudomonas but no other genera? And Pseudomonas remained dominant on day 317 at 26.9%. Why the relative abundance of Pseudomonas decreased after day 271? Any credible reasons behind this phenomena? Please explain. It is very interesting.
Q7. P10, L-298-299, “Pseudomonas may have influenced the degradation of diesel oil during winter in the present study’.
The discussion part clearly indicates that all the similar studies about the relative abundance of Pseudomonas [references 36-40] conducted at 4°C. In the present study what was the temperature? And briefly explain what were the precautions/challenges initiated while performing this research?
Q8. P11, L-333-334, “In the present study, seasonal rhizoremediation performance was investigated using tall fescue in diesel oil-contaminated soil”.
What are the other plants (instead of tall fescue plants) which can be substituted for tall fescue in this study? How can we extend this study to other regions of the world, precisely at higher temperatures or under numerous other environmental conditions?
Author Response
III. Reviewer #3
This paper investigates the cyclic characteristics of rhizoremediation in Petroleum PH-contaminated wetlands and the role of petroleum hydrocarbon (PH)-degrading bacteria such as Genus Pseudomonas, genera Immundisolibacter and Lysobacter studied in soil contaminated with diesel oil. Tall fescue seedlings were planted in the contaminated soil and rhizoremediation performance was monitored for 317 days. The scientific content of the manuscript is very sensational and significant for the bioremediation of soil pollutants and rejuvenation of the PH-contaminated sites using various bacterial genera. The paper reports the solution for a contemporary issue and worth publishable. The manuscript is very well organized and written with ample references and adequate for publication after making some minor corrections described below.
===============
(1) P2, L-50-55, “Tall fescue grows well under a wide range of environmental stressors, including strongly acid to alkaline conditions, and its deep root system allows it to thrive in wetlands and other low-lying sites with high moisture. Tall fescue is an introduced grass that is common in Pacific Northwest wetlands and has been widely found in reclaimed tidal lands along the mid-west coast of South Korea”. Please correct the English grammar.
=> We have corrected the English grammar in the sentence (Page 2, line 54).
(2) P2, L-78-79, “Finally, the association between rhizoremediation performance, environmental factors, and the bacterial community was comprehensively assessed.” Please correct the grammar of the sentence.
=> We have corrected the sentence (Page 2, lines 78–80).
(3) P4, L-139-141, “The association between rhizoremediation performance, environmental factors, and bacterial genera was analyzed using extended local similarity analysis (eLSA) and visualized with Cytoscape.” Please correct the English grammar.
=> We have corrected the sentence (Page 4, lines 144–146).
(4) P4, L-170-172, “The TPH concentration subsequently did not change significantly, with an average concentration and removal efficiency of 5,970–7,535 mg-TPH·kg-soil-1 and 69.2–75.6%, respectively, from days 84 to 152.” Please correct the sentence.
=> We have corrected the sentence (Page 5, lines 176–179).
(5) P6, L-195-199, “Similar to these results, the TPH removal rate in the present study ranged from 69.2% to 75.6% for days 61–152. However, after re-contamination with diesel on day 153, the TPH removal rate declined to 30.1–37.5%. This might be because the bacterial activity for TPH degradation in the tall fescue rhizosphere was disrupted as the temperature fell after the re-contamination”.
Here the author claims that after re-contamination with diesel, the bacterial activity for TPH degradation in the tall fescue rhizosphere was disrupted. What are the other plausible reasons for the declination of TPH removal rate? Is this happen only due to temperature? What about water content and organic matter content? Also please correct the sentence structure.
=> We much appreciated this valuable comment. There are various influencing factors on PH degradation, such as temperature, soil pH, soil texture, and water content, et cetera. It has been additionally discussed in the revised manuscript (Pages 6–7, lines 206–213 & lines 223–237). Moreover, the increase of the TPH concentration in the soil due to re-contamination might have an impact on the bacterial activity for TPH removal. Similar results have been previously reported, and it is described in the manuscript (Pages 6–7, lines 214–222).
(6) P7, L-238-241, “On day 210, which was during winter, the relative abundance of Pseudomonas increased to 15.7%, followed by Sphingomonas (11.9%) and Massilia (9.5%). On day 271, the relative abundance of Pseudomonas increased considerably to 55.7%, whereas the other genera were all below 8%”.
According to the experimental procedure narrated in Page 3; L99, the day 210 (December 23rd, 2020) and day 271 (February 22nd, 2021), which is in the peak of winter and the data shows that On day 271, the relative abundance of Pseudomonas increased considerably to 55.7%. Could you please explain why this is happened to only Pseudomonas but no other genera? And Pseudomonas remained dominant on day 317 at 26.9%. Why the relative abundance of Pseudomonas decreased after day 271? Any credible reasons behind this phenomena? Please explain. It is very interesting.
=> Only the increase in relative abundance of Pseudomonas was described in the manuscript because its change was particularly remarkable in the community structures. In fact, however, the relative abundance of Alkanindiges (7.4% on day 271) and Massilia (9.5% on day 210) also increased during winter. It is very difficult to suppose which factors had an impact on the change in Pseudomonas in winter, because the contribution factors vary widely. In general, viscosity of PH increases and solubility decreases at low temperatures, resulting in delay of biodegradation [1]. Therefore, the production of biosurfactant or bio-emulsifiers plays an important role in PH degradation at low temperatures [1]. It is generally reported that Pseudomonas is capable of producing biosurfactant (rhamnolipid-type) [2]. In addition, Malavenda et al. (2015) reported the biosurfactant production by Pseudomonas at low temperatures of 4–15°C [2]. Although there have been very few studies on biosurfactant production by Pseudomonas at low temperatures, it might have had a positive effect on TPH degradation. We have additionally discussed this explanation in the revised manuscript (Page 10, lines 330–335).
- Varjani, S.J.; Upasani, V.N. A new look on factors affecting microbial degradation of petroleum hydrocarbon pollutants. Biodeterior. Biodegrad. 2017, 120, 71-83.
- Abouseoud, M.; Maachi, R.; Amrane, A. Biosurfactant production from olive oil by Pseudomonas Commun. Curr. Res. Educ. Top. Trends Appl. Microbiol. 2007, 340–347.
- Malavenda, R.; Rizzo, C.; Michaud, L.; Gerce, B.; Bruni, V.; Syldatk, C.; Hausmann, R.; Giudice, A.L. Biosurfactant production by Arctic and Antarctic bacteria growing on hydrocarbons. Polar 2015, 38, 1565-1574.
(7) P10, L-298-299, “Pseudomonas may have influenced the degradation of diesel oil during winter in the present study’.
The discussion part clearly indicates that all the similar studies about the relative abundance of Pseudomonas [references 36-40] conducted at 4°C. In the present study what was the temperature? And briefly explain what were the precautions/challenges initiated while performing this research?
=> Unfortunately, temperature in soil was not measured in this study. Instead, the ambient temperature is shown in Figure 1a. We divided the experimental period into four sub-periods based on the average ambient temperature. The average ambient temperature for each period was 24.2°C for summer, 14.3°C for autumn, 0.2°C for winter, and 10.2°C for spring (Page 4, lines 152–156).
In this study, dynamics of soil bacterial community were explored during rhizoremediation of diesel-contaminate soil cultivated with tall fescue. It is considered important to maintain plant biomass which plays a critical role in rhizoremediation. Therefore, various soil amendment such as compost, manure, sewage sludge, and organic wastes can be amended to enhance plant growth and bacterial abundance in the rhizosphere [4–6]. In addition, plant growth promoting bacteria (PGPB), which strengthens plant growth, can also be inoculated to enhance rhizoremediation performance [7,8].
Although this study investigated rhizoremediation using plant (tall fescue) in diesel-contaminated soil, observation of plants could not be conducted because the experiment should be continuously monitored for a year on a limited scale. Therefore, in a future work, plants observation would be conducted through a destructive method or expansion of the experimental scale. It also plans to further analyze the plant growth promoting (PGP) ability (such as indole-3-acetic acid (IAA) production and ACC deaminase) of soil bacterial communities.
- Adams, G.O.; Fufeyin, P.T.; Okoro, S.E.; Ehinomen, I. Bioremediation, biostimulation and bioaugmentation: a review. J. Environ. Bioremediation Biodegrad. 2015, 3, 28-39.
- Namkoong, W.; Hwang, E.Y.; Park, J.S.; Choi, J.Y. Bioremediation of diesel-contaminated soil with composting. Pollut. 2002, 119, 23-31.
- Hussain, F.; Hussain, I.; Khan, A.H.A.; Muhammad, Y.S.; Iqbal, M.; Soja, G.; Reichenauer, T.G.; Yousaf, S. Combined application of biochar, compost, and bacterial consortia with Italian tyegrass enhanced phytoremediation of petroleum hydrocarbon contaminated soil. Exp. Bot. 2018, 153, 80-88.
- Mhatrea P.H.; Karthikb, C.; Kadirvelu, K.; Divya, K.L.; Venkatasalam, E.P.; Srinivasa, S.; Ramkumar, G.; Saranya, C.; Shanmuganathan, R. Plant growth promoting rhizobacteria (PGPR): a potential alternative tool for nematodes bio-control. Agric. Biotechnol. 2019, 17, 119-128.
- Gopalakrishnan, S.; Sathya, A.; Vijayabhararhi, R.; Varshney, R.K.; Gowda, C.L.L.; Krishnamurthy, L. Plant growth promoting rhizobia: challenges and opportunities. 3 Biotech, 2015, 5, 355-377.
(8) P11, L-333-334, “In the present study, seasonal rhizoremediation performance was investigated using tall fescue in diesel oil-contaminated soil”.
What are the other plants (instead of tall fescue plants) which can be substituted for tall fescue in this study? How can we extend this study to other regions of the world, precisely at higher temperatures or under numerous other environmental conditions?
=> In general, contaminant-tolerant plant species with well-developed root systems are often used to promote rhizoremediation [9]. Several plant species such as maize, tall fescue, ryegrass, and alfalfa have been widely utilized for rhizoremediation of oil-contaminated soils [10]. Among these plant species, a suitable (or native) plant which can adapt well to the local environment should be selected due to their adaptive capacities to climate, geological and environmental conditions.
- Baoune, H.; Apariciio, J.D.; Acuña, A.; Hadj-Khelil, A.O.; Sanchez, L.; Polti, M.A.; Alvarez, A. Effectiveness of the Zea mays-Streptomyces association for the phytoremediation of petroleum hydrocarbons impacted soils. Environ. Saf. 2019, 184, 109591.
- Hussain, I.; Puschenreiter, M.; Gerhard, S.; Schöftner, P.; Yousaf, S.; Wang, A.; Syed, J.H.; Reichenauer, T.G. Rhizoremediation of petroleum hydrocarbon-contaminated soils: improvement opportunities and field applications. Exp. Bot. 2018, 147 202-219.
Reviewer 4 Report
- The authors are encouraged to present a critical literature review showing the gap in the subject clearly.
- The authors may more explain the necessity of this research and the objectives.
- Explain the assumptions and limitations of the present study.
- Discussion section needs more physical explanation rather than a descriptive presentation. A statistical analysis may enrich the presentation of results. In addition, a comparison with some existing studies is required.
- Conclusion section can better reveal the application of this study.
- What is the impact of this study on the cost of environmental projects?
- How can one extend the results? What are specific suggestions for the future work?
Author Response
- Reviewer #4
The authors are encouraged to present a critical literature review showing the gap in the subject clearly. The authors may more explain the necessity of this research and the objectives. Explain the assumptions and limitations of the present study. Discussion section needs more physical explanation rather than a descriptive presentation. A statistical analysis may enrich the presentation of results. In addition, a comparison with some existing studies is required. Conclusion section can better reveal the application of this study. What is the impact of this study on the cost of environmental projects? How can one extend the results? What are specific suggestions for the future work?
=> We much appreciated the precious comments. A statistical analysis (One-way ANOVA and multiple comparison) has been carried out, and it was added in the revised manuscript (Page 4, lines 146–148; Page 5, lines 172–185; Page 6, Figure 2). In addition, we have rewritten Discussion and Conclusion in the revised manuscript as the recommend (Page 6, lines 206–213; Page 7, lines 223–237; Page 10, lines 330–335; Pages 11–12, lines 369–391).
Reviewer 5 Report
Dear authors, it was interesting for me to read your article. The direction of your research seems relevant. I hope my comments will be useful for further improvement of their quality.
- The main complaint to the manuscript is the unsatisfactory design of the experiment. During the experiment, the soil microbiota is exposed to the simultaneous action of several factors, the effect of which cannot be separated by statistical methods. The first factor is seasonal changes in temperature and rainfall. The second factor is that adding diesel oil to the soil triggers ecological succession. In the course of succession, species replace each other, the dominant positions are successively occupied by microorganisms with different trophic needs. The third factor is the addition of a second portion of diesel oil in the middle of the experiment, which significantly changed the concentration of the pollutant in the soil, which should have affected the soil bacteria. It is impossible to understand what contribution each of these factors made to the taxonomic diversity of bacteria. Since it is too late to correct this error at the moment, at least the discussion of the results should be rewritten so that climatic factors are not considered as the only cause of changes in species composition.
- For the article devoted to rhizoremediation, little attention is paid to plants. Although the purpose of the article was not to study plants, it is necessary to include a little information about them. What age seedlings were planted in the soil? At what rate did the mass of roots increase? Root systems begin to influence remediation only after reaching the minimum critical volume. Did the plants actively grow during the entire period of the experiment or did they go dormant?
- Complete the soil and sampling information. What kind of microbial inoculum was in the compost? From what depth were soil samples taken? Soil samples were taken from the surface of the roots or not?
4. Fig. 3 - not all color designations are deciphered in the legend.
Author Response
The main complaint to the manuscript is the unsatisfactory design of the experiment. During the experiment, the soil microbiota is exposed to the simultaneous action of several factors, the effect of which cannot be separated by statistical methods. The first factor is seasonal changes in temperature and rainfall. The second factor is that adding diesel oil to the soil triggers ecological succession. In the course of succession, species replace each other, the dominant positions are successively occupied by microorganisms with different trophic needs. The third factor is the addition of a second portion of diesel oil in the middle of the experiment, which significantly changed the concentration of the pollutant in the soil, which should have affected the soil bacteria. It is impossible to understand what contribution each of these factors made to the taxonomic diversity of bacteria. Since it is too late to correct this error at the moment, at least the discussion of the results should be rewritten so that climatic factors are not considered as the only cause of changes in species composition.
=> We much appreciated the precious comments. The purpose of this study was to examine PHs degradation and bacterial community dynamics during rhizoremediation under a natural climatic conditions. Accordingly, seasonal changes in temperature and rainfall were not artificially controlled. In addition, soil was amended with diesel in this study because the study examined the “dynamics of bacterial communities in diesel-contaminated soil”, even though the addition of diesel could trigger ecological succession. Moreover, after approximately 70% of TPH was removed on day 61, there was no significant change in TPH removal rate before re-contamination of diesel (day 153). The bacterial community structure was also similar for days 61–117 before re-contamination as shown in Figures 3 and 4. Therefore, re-contamination of diesel was carried out even though it could cause changes in bacterial communities again.
As your comments, climatic factors are not considered as the only cause of changes in species composition. Accordingly, we authors rewritten the manuscript (Page 6, lines 206–213; Page 7, lines 223–237), and much appreciated for pointing this out. However, in this study, only temperature, soil pH, soil water and organic matter contents were measured as the environmental factors, so there was a limit to explaining other factors affecting rhizoremediation and bacterial community dynamics. Therefore, in further studies, not only rhizoremediation and bacterial community dynamics, but also soil enzyme activity (such as dehydrogenase, indole-3-acetic acid (IAA) production, and urease activity, et cetera) and plant biomass would be comprehensively analyzed. This limitation and further suggestions have been added in the revised manuscript (Page 12, lines 384–387).
For the article devoted to rhizoremediation, little attention is paid to plants. Although the purpose of the article was not to study plants, it is necessary to include a little information about them. What age seedlings were planted in the soil? At what rate did the mass of roots increase? Root systems begin to influence remediation only after reaching the minimum critical volume. Did the plants actively grow during the entire period of the experiment or did they go dormant?
=> We have much appreciated this valuable comment. Tall fescue seeds were sown for 2 months in non-contaminated soil. After 2 months, 100 tall fescue seedlings were planted in the pot. This description has been added in the revised manuscript (Page 3, lines 94–97).
In this study, plants could not be obtained because it was continuously monitored for a year in the limited-scale pot. Therefore, change in the root mass of tall fescue could not be observed. However, tall fescue seedlings were transferred to the pot after 2 months of sufficient growth in non-contaminated soil, thus it is thought that it reached the minimum critical volume for rhizoremediation. Additionally, tall fescue actively grew well during the experimental period, particularly in summer and autumn (as shown in Figure A below). However, the growth of tall fescue was inactive after re-contamination of diesel on day 153, which was thought to have affected the decrease in TPH removal rate after re-contamination.
Figure A. Growth of tall fescues used in the study on (a) day 0 (May 27th, 2020) and (b) 117 (September 21st, 2020).
Complete the soil and sampling information. What kind of microbial inoculum was in the compost? From what depth were soil samples taken? Soil samples were taken from the surface of the roots or not?
=> The bacterial community structure in compost was not investigated in present study. However, in our previous report, the bacterial community structures were compared as compost addition in diesel-contaminated and tall fescue-cultivated soil [11]. The genus Acidicapsa exhibited the highest abundance (11.1%) in the soils without compost treatment, followed by Gemmatirosa (7.8%) and Gemmata (6.7%). In contrast, Stenotrophomonas (6.8%), Effusibacillus (5.9%), and Gemmatirosa (4.9%) were relatively abundant in compost-amended soil [11]. Through this result, it is possible to estimate the effect of compost on the bacterial community structures during rhizoremediation.
Since tall fescue should be maintained in the pot for a year without any damage, soil samples could not collected from the surface of the roots. Instead, soil samples were collected at a depth of 5–15 cm as close to the roots as possible. The sampling method of soil have described in the revised manuscript (Page 3, lines 98–99).
- Lee, Y-Y.; Seo, Y.; Ha, M.; Lee, J.; Yang, H.; Cho, K-S. Evaluation of rhizoremediation and methane emission in diesel-contaminated soil cultivated with tall fescue (Festuca arundinacea). Res. 2021, 194, 110606.
Fig. 3 - not all color designations are deciphered in the legend.
=> We have much appreciated this valuable comment. We have changed Figure 3 in the revised manuscript.

Round 2
Reviewer 4 Report
The authors replied my comments and I recommend the publication of this manuscript.
Reviewer 5 Report
Dear authors, thank you for the detailed answers. I hope you will be able to carry out your further research plans. Nevertheless, I will advise in further experiments to make new pots with the same concentration of the pollutant instead of re-polluting old pots, if it is necessary to study the active phase of bioremediation. Then it will be possible to exclude the factor of different pollution strength.